# Spectrally Selective Detection of Short Spin Waves in Magnetoplasmonic Nanostructures via the Magneto-Optical Intensity Effect

**DOI:** 10.3390/nano12030405

**Published:** 2022-01-26

**Authors:** Olga V. Borovkova, Saveliy V. Lutsenko, Mikhail A. Kozhaev, Andrey N. Kalish, Vladimir I. Belotelov

**Affiliations:** 1Russian Quantum Center, Novaya Str. 100, Skolkovo, 143025 Moscow, Russia; savlucenko@yandex.ru (S.V.L.); mikhail.kozhaev@gmail.com (M.A.K.); kalish@physics.msu.ru (A.N.K.); v.belotelov@rqc.ru (V.I.B.); 2Faculty of Physics, Lomonosov Moscow State University, Leninskie Gory, 119991 Moscow, Russia; 3NTI Center for Quantum Communications, National University of Science and Technology MISiS, Leninsky Prospekt 4, 119049 Moscow, Russia

**Keywords:** magnetooptics, spin waves, surface plasmon polaritons

## Abstract

A method of spectrally selective detection of short spin waves (or magnons) by means of the transverse magneto-optical (MO) intensity effect in transmission in the magnetoplasmonic nanostructure is proposed. We considered the spin waves with a wavelength equal to or less than (by an integer number of times) the period of the plasmonic structure, that is, of the order of hundreds of nanometers or 1–2 μm. The method is based on the analysis of the MO effect spectrum versus the modulation of the sample magnetization (created by the spin wave) and related spatial symmetry breaking in the magnetic layer. The spatial symmetry breaking leads to the appearance of the MO effect modulation at the normal incidence of light in the spectral range of the optical states (the SPP and the waveguide modes) and the breaking of the antisymmetry of the effect with respect to the sign of the incidence angle of light. We reveal that the magnitude of the MO effect varies periodically depending on the spatial shift of the spin wave with respect to the plasmonic grating. The period of this modulation is equal to the period of the spin wave. All these facts allow for the detection of spin waves of a certain wavelength propagating in a nanostructure by measuring the MO response.

## 1. Introduction

Recently, an interest in spin waves, or magnons, has raised both from the side of fundamental science and from the point of view of practical applications due to a variety of the possibilities that are offered by magnonics for the data storage and processing systems [1,2,3,4,5,6,7]. Magnonic devices would have much lower power consumption and operate at frequencies up to dozens of THz.

However, currently, novel materials and magnetic nanostructures where the excitation, control and detection of spin waves can be realized are required. In recent years, optical techniques for magnonics have been actively developing. Optomagnonics provides several approaches for spin waves generation, such as the photomagnetic effect and the inverse magneto-optical (MO) effects [5,6,7,8,9]. These methods are more flexible in comparison with the nanoantennas that can also serve as a sources of the spin waves.

Regarding spin wave detection, one should mention the Brillouin light scattering (BLS) spectroscopy and μBLS technique [10,11], where the scattering of the photons by spin waves provide the magnons propagation image. This method provides ample opportunities but it requires expensive hi-tech equipment including a Fabry–Perot interferometer. Another way to detect the spin waves is to measure the Faraday or Kerr rotation angle [3,5,7,12,13,14,15] of the ferromagnetic layer where the spin waves are excited. However, this method provides a detection of spin waves in a wide frequency range.

Therefore, it is difficult to detect the spin waves produced by the certain source, such as nanoantenna whose dimensions determine the wavelength of the spin wave. The existent methods limit the possibilities to deal with the narrowband magnonic signals.

To solve this problem, in this paper, we propose employing magnetoplasmonic nanostructures that make it possible to detect spin waves in a narrow frequency range by means of the MO intensity effect in transmission enhanced nearby the optical resonances related to the excitation of the surface plasmon polaritons (SPP) and the waveguide modes.

The magnetic nanostructures are well-known for their ability to enhance the MO effects or to amplify and enrich their properties [16,17,18,19,20,21,22]. For instance, it was recently shown that the spatial symmetry breaking in the magnetoplasmonic nanostructures can lead to the novel MO effect, the transverse intensity magneto-optical effect in transmission [23].

One of the features of the reported phenomenon was the fact that the MO effect is nonzero at the normal incidence of light. In Ref. [23], the spatial symmetry breaking was created due to the asymmetry of the unit cell geometry in the plasmonic grating. The excitation of the spin wave or the waves of magnetization in a continuous magnetic medium also acts as a spatial symmetry breaking in the nanostructure. However, the MO intensity effect turns to zero under certain conditions despite the spatial symmetry breaking made by the spin wave.

Here, we address the case where the period of the plasmonic grating is equal to the integer number of the magnetization oscillations periods (or periods of the spin wave), and the spatial symmetry becomes broken inside one grating cell. Thus, we are interested in the detection of the short spin waves, i.e., the spin waves with the wavelength from several hundreds of nanometers to 1–2 μm.

In this paper, we analyze how the magnetization modulation influences the MO intensity effect in transmission near the spectral range of the SPP and the waveguide modes. Based on this, a novel approach for the detection of the short spin waves of the narrow frequency range by means of the measurements of the MO effect in the nanostructure is proposed.

## 2. Magnetization Modulation Due to the Spin Waves Excitation

Magnetization modulation in a ferrimagnetic dielectric material can be created in different ways. Among them, there are the alternating magnetic domains [24,25], external mechanical stress or stretching [26,27] and the excitation of spin waves [28,29]. In this paper, we are interested in the latter option. When the spin waves propagate in a magnetized ferrimagnetic dielectric medium, the magnetic dipole moments deviate from the direction given by the external magnetic field. As a result, the magnetization of the ferrimagnetic dielectric becomes modulated along a certain direction. In this section, we consider how the oscillating components of the magnetization vector will be directed depending on the spatial coordinate. Further analysis will be based on the obtained spatial distribution of the magnetization vector.

We address here the volume magnetostatic spin waves, both forward and backward [28,29]. Such waves can be excited either in a perpendicularly magnetized ferrimagnetic sample (forward volume spin waves) or when the external magnetic field is in the plane of the ferrimagnetic film (backward volume spin waves). Spin waves correspond to the oscillation of the magnetization component in the plane perpendicular to the magnetic field. The dispersion properties of the analyzed magnetic waves have been thoroughly studied in different magnetic materials, such as yttrium iron garnet [30] or bismuth-substituted yttrium iron garnet [31]. Thus, it is necessary to understand how the propagating spin waves change the optical properties of the material via the magnetization variation.

For this purpose, we considered the constant magnetic field $H_{0}$ directed along *y*-axis as it is shown in Figure 1. As a result, the magnetization vector **M** precession occurs in the *xz*-plane. Therefore, two oscillating components of the magnetization, $m_{z}$ and $m_{x}$, appear.

Similarly, when the constant external magnetic field is directed along the *z*-axis, perpendicular to the surface of the ferrimagnetic sample, the corresponding oscillating components of the magnetization are $m_{y}$ and $m_{x}$.

We consider plane spin waves that have the harmonic dependence on the spatial coordinate, in our particular case, y. Thus, the components of the oscillating magnetization component are proportional to exp(−ikr) or, in the case under consideration, to $\exp(-ik_{\mathrm{SW}}y)$. The same harmonic dependence is typical for the oscillating components of the magnetization, in particular, for $m_{x}$ (Figure 1).

It should be noted that, although for both types of the volume magnetostatic spin waves, there are two oscillating components of the magnetic field, here we take into account only the $m_{x}$ component of the magnetization due to the fact that we address the intensity MO effect that is insensitive to the polar ($m_{z}$) and longitudinal ($m_{y}$) components of the magnetization.

## 3. The Magnetoplasmonic Nanostructure with the Magnetization Modulation

Here, we address the magnetoplasmonic nanostructures composed of a thin layer (or a film) of a magnetic material deposited on the surface of a nonmagnetic dielectric substrate of gadolinium gallium garnet (GGG). The top surface of the magnetic layer is covered by a one-dimensional plasmonic grating made of noble metal. A ferrimagnetic dielectric of bismuth-substituted yttrium iron garnet (BIG) is chosen as a magnetic layer due to its transparency in the visible spectral range and the significant MO response [15,32,33,34].

We assume that the constant external magnetic field $H_{0}$ can be applied along the *y*-axis (see Figure 2) or along the *z*-axis (not shown), and the magnetization saturation of the ferrimagnetic film is achieved. In the ferrimagnetic dielectric BIG films, the spin waves can be excited by means of either stripe nanoantennas [35,36] or due to a combination of short optical pulses and the inverse Faraday effect [31].

When the plasmonic nanostructure is illuminated by the plane wave of *p*-polarized light (the vector E lies in plane zOy as it is shown in Figure 2), the SPP wave is excited [17]. Strictly speaking, there are excited two SPP modes propagating along the *y*-axis forward and backward. The corresponding spatial intensity distribution of the SPP wave is shown in the ferrimagnetic layer by a yellow-red-black color plot. This is given in arbitrary units and is normalized to the peak value.

Along with the optical light distribution, the ferrimagnetic layer experiences the modulation of dielectric properties due to the propagation of the spin wave and the related magnetization modulation. When the spin waves propagate in the plane of the ferrimagnetic film perpendicular to the plasmonic grating axis (as shown in Figure 2), they create the oscillating component of the magnetization along the *x*-axis (see also Figure 1).

The direction of a magnetization modulation coincides with the direction of modulation of a one-dimensional plasmonic grating (see Figure 2). We describe the maximum magnetization modulation due to the oscillations of the magnetic field $h_{x}$ in the ferrimagnetic layer as a modulation of the local gyration of the material
(1)$$g\left(y\right)=g_{0}\sin\left(k_{SW}y+\phi \right),$$
where *g* is related to magnetization-induced non-diagonal components of the dielectric tensor ($g=i\varepsilon_{yz}=-i\varepsilon_{zy}$), $g_{0}$ is the amplitude of the magnetization modulation, $k_{SW}$ is the wavevector of the spin wave, $k_{SW}=2\pi /\lambda_{SW}$, and ϕ is a phase shift between the spin wave and the plasmonic grating. Such a phase shift corresponds to the spatial shift between ‘zero’ of the gyration modulation and the left edge of the gold grating, $\phi =k_{SW}\Delta y$ (see Figure 2). Parameter Δy can take any values in the interval [0,P), where *P* is a period of the plasmonic grating.

The right boundary of the interval is punctured, as it already belongs to the next period. Thus, the phase ϕ lays in the interval $\left[0,2\pi P/\lambda_{SW}\right)$. Due to the periodicity of the magnetization in the ferrimagnetic layer, the total phase $k_{SW}y+\phi$ changes inside the same interval $\left[0,2\pi P/\lambda_{SW}\right)$. The magnetization distribution in the sample given by Equation (Equation 1) can be created by exciting spin waves (magnons), in this case, ϕ changes in time as $\phi =\omega_{SW}t$, $\omega_{SW}$ is the spin wave frequency, and *t* is time.

Since $\omega_{SW}$ is of the terahertz range and therefore is much less than the optical frequency. For the determination of the optical and MO response, we can consider the static magnetization distribution given by Equation (Equation 1) for each moment in time. If the wavelength of the spin wave is equal to the integer number of the periods of the plasmonic grating, the MO effects observed in such nanostructures have some peculiarities that can be used for the detection of magnons.

We emphasize that the described method will not detect all the spin waves propagating in the ferrimagnetic layer but only the magnons of a certain wavelength. This causes the spectral selectivity of the proposed approach. Moreover, the requirement of the coincidence of the plasmonic grating and spin wave period means that the proposed method can be effectively employed for the detection of the short spin waves with a wavelength of less than 2 μm. The greater periods of the plasmonic grating will not provide effective excitation of the SPP waves, and the detection scheme should be changed.

To explore the intensity MO effect in transmission, the nanostructure is illuminated by the plane wave of *p*-polarized light as shown in Figure 2. We vary the wavelength and the incidence angle of the input light and analyze the transmission spectrum of the structure due to the variation of the oscillating component of magnetization $m_{x}$.

## 4. The Intensity MO Effect in the Magnetoplasmonic Nanostructure with the Magnetization Modulation

The transverse magneto-optical intensity effect in transmission [23] is determined as a relative change of the transmitted light intensity $T(m_{x})$ when the structure is re-magnetized. In our case, this means that the effect is measured for two opposite directions of $m_{x}$
(2)$$\delta_{T}=2\frac{T\left(m_{x}\right)-T\left(-m_{x}\right)}{T\left(m_{x}\right)+T\left(-m_{x}\right)}.$$

If one considers the *p*-polarized light falling down the magnetic material with spatial modulation of the gyration (Equation 1), it is easy to show that the relative change of the transmitted light [37,38] is proportional to the gyration *g* of the magnetic material. For the considered system, this indicates that
(3)$$\delta_{T}∼\sin\left(k_{SW}y+\phi \right).$$

In the absence of the plasmonic grating, the phase shift ϕ is undetermined, and, for the input plane wave, the integral over the *y*-coordinate results in the opposing impact from positive and negative half periods of the sine function. However, in the case of the magnetoplasmonic nanostructure, as in Figure 2, two aspects become crucial.


The coincidence of the period of the plasmonic nanostructure and the magnetization modulation. When the periods are different, destructive interference occurs, causing the overall MO response to be distorted. Thus, we limit our consideration by $n\lambda_{SW}=P$, where n∈Z, the period of the plasmonic nanostructure is equal to integer number, *n*, of magnon wavelength.From Equation (Equation 3) one can see that the MO effect depends on the phase shift ϕ or, equally, on Δy. By varying the spatial shift Δy between the plasmonic grating and magnetization modulation, the MO effect can be controlled and even turned to zero, when $k_{SW}y+\phi =0$ or $k_{SW}y+\phi =\pi$. As mentioned above, the plasmonic grating period should be equal to an integer number of magnon wavelength, i.e., $P=n\lambda_{SW}$. It can be easily shown that the phase $k_{SW}y+\phi$ takes the values 0 and π for 2*n* times in the interval $\left[0,2\pi P/\lambda_{SW}\right)$.


Thus, the magnetoplasmonic nanostructure with spatial symmetry breaking can have zero MO effect under certain conditions. The dependence on the phase shift will be discussed in further detail. To clear up the impact of the spin waves on the MO effects in the magnetoplasmonic nanostructure, we analyze the wavelength and angular-resolved transmission and $\delta_{T}$ spectra of the addressed nanostructures of different periods and compare the results with the case of the uniform magnetization of the ferrimagnetic film.

We addressed the MO properties by means of the numerical simulation of the periodic magnetoplasmonic nanostructures by the rigorous coupled-wave analysis (RCWA) [39,40]. This method is an appropriate tool for simulation of the optical and magneto-optical response of multilayered structures with lateral 1D or 2D periodicity. The first step of the method is the solution of Maxwell’s equations in truncated Fourier space so that Bloch waves are found as the solution of the eigenwaves problem. Then, at the second step, the boundary conditions for interlayer interfaces are applied to obtain the algebraic set of equations for transmission and reflection coefficients. The RCWA method provides calculation of both far-field characteristics (such as the intensity and polarization of scattered waves for all diffraction orders) and near-field distribution of electromagnetic field components.

Two sets of parameters of the plasmonic grating were considered. The first set was chosen to provide excitation of the surface plasmon polariton (SPP) modes that occurs in the spectral range of 650–900 nm, and the second configuration supports the excitation of the waveguide modes in the magnetic layer at wavelengths of 550–625 nm.

For the excitation of the SPP modes, the period of the plasmonic grating is chosen to be 630 nm, and the width of the air gap is 85 nm; therefore, 86.5% of the surface is covered by gold. The thickness of the metal is 80 nm, and the thickness of the ferrimagnetic film is 100 nm. Note that such a thin ferrimagnetic layer prevents excitation of the waveguide modes, and for them a slightly different configuration was chosen.

We calculated the angular and wavelength-resolved transmission and MO effect spectra. Although the normal incidence of light is of the most interesting in the context of the asymmetric magnetoplasmonic nanostructures, the spectra were calculated also for oblique light, as shown in Figure 2, to facilitate the analysis and to make the SPP resonances easily distinguishable. The input light angle is given in Figure 3 in degrees and was measured from the normal to the sample plane.

Figure 3a,d show the transmission and $\delta_{T}$ spectra of the nanostructure with uniform magnetization of the ferrimagnetic layer. The typical resonant peculiarities related to the excitation of the SPP modes with the orders m=±2 (upper branch) and m=±3 (lower branch) are denoted in the left column of Figure 3. The transmission spectrum is symmetric, and the spectrum of the magneto-optical effect is antisymmetric due to the change of the sign of the input light angle and the corresponding change of the spatial symmetry conditions. Due to the symmetry, the $\delta_{T}$ spectrum is equal to zero at the normal incidence of light.

When the magnetization of the ferrimagnetic film is modulated due to the excitation of the spin waves, it causes changes in both the transmission and $\delta_{T}$ spectra. In the center column of Figure 3, the corresponding optical and magneto-optical spectra are given for the cases when the magnetization modulation period is equal to the period of the plasmonic grating. The phase shift was chosen to provide the maximum value of $\delta_{T}$.

One can see that in the spectral range of the SPP modes of the orders m=±2 (in particular, near λ=850 nm) the MO effect has a constant sign, and its value is almost the same over the wide range of the incident angles. Thus, the MO effect is nonzero at the normal incidence of light in the spectral range of the SPP modes of the orders m=±2. The SPP modes of the orders m=±3 (near λ=700 nm) also reveal the nonzero MO effect at the normal incidence of light. Therefore, the spatial symmetry breaking caused by the magnetization modulation makes the changes in the MO effect offering the method to detect the presence of the spin wave. The only problem is that the magnitude of $\delta_{T}$ is 10-times lower than for the case when the magnetization is unmodulated (see Figure 3d).

This problem can be overcome when the period of the magnetization modulation is equal to the half of the plasmonic grating period. The reason is that, in the considered setting at the interface of the [plasmonic grating]/[ferrimagnetic dielectric] for λ=850 nm in the air, the SPP wavelength is about 313 nm [17]. Thus, if $\lambda_{SW}=P/2$, the SPP mode wavelength $\lambda_{SPP}$ is equal to $\lambda_{SW}$, and the impact of the spatial symmetry breaking created by the magnetization modulation is stronger. The corresponding transmission and $\delta_{T}$ spectra are given in the right column of Figure 3 for the case of zero phase shift. The magnitude of the MO effect is comparable with the case of uniform magnetization distribution.

Now, let us analyze the dependence of the MO effect $\delta_{T}$ on the phase shift, ϕ, and how this can help to detect the spin waves in the magnetic layer. In Figure 4, the spatial distributions of the $|H_{x}{|}^{2}$ component of the excited SPP wave and the magnetization modulation with two different phase values are shown. In Figure 4a, the spatial distribution of the magnetization is non-symmetric inside the plasmonic grating cell, but the impacts of the half periods of the magnetization to the magneto-optical response compensate for each other.

This makes the whole nanostructure insensitive to the re-magnetization, and, although the SPP modes of orders m=±2 are excited in the nanostructure, the corresponding resonances do not emerge in the $\delta_{T}$ spectrum (see Figure 3f). On the contrary, the space shift between the plasmonic grating and the magnetization modulation given in Figure 4b breaks the spatial symmetry and provides the magneto-optical response for the same magnon wavelength. These two cases provide different MO effect values. We propose to employ the harmonic dependence of the MO effect on the magnetization modulation for the purposes of spin wave detection.

As the spin wave propagates in the ferrimagnetic film, its location and spatial displacement relative to the plasmonic structure changes. We revealed that the transmittance magnitude as well as the value of the MO effect change depending on that spatial displacement or phase shift between the plasmonic grating and the spin wave. Moreover, the dependence observed for the MO effect is periodic with a period equal to the modulation period of the magnetization. This is shown in Figure 5 where the spectra of the MO effect versus the phase shift ϕ for two different wavelengths of the spin wave are given. The phase shift is given in radians to avoid confusion with the input light angle measured in degrees earlier.

The spectra for two periods of the magnetization modulation are compared. In Figure 5a, the period of the magnetization modulation is equal to the plasmonic grating period, while in Figure 5b, the magnetization modulation period is two-times smaller. In addition, the spectral dependence of the largest value of the MO effect for different periods of magnetization modulation may turn out to be more convenient for experimental measurements. The corresponding graph is shown in Figure 5c. One can see that, when the plasmonic grating period is two-times greater than the period of the spin wave modulation, the MO effect is stronger than in case of the period coincidence.

The revealed phenomenon opens up the novel possibilities to control the magnitude of the effect and, on the other hand, to detect the spin waves in the magnetic layer of the nanostructure. We are not limited by the precise coincidence of the period of the plasmonic grating and the wavelength of the spin waves. On the contrary, the higher frequencies of the magnons can be easily detected and distinguished by measuring the time dependence of the $\delta_{T}$ spectrum (i.e., a phase shift).

In addition to the SPP modes, the proposed method can be implemented by the excitation of the waveguide modes in the magnetic layer. For this purpose, a slightly different design of the magnetoplasmonic nanostructure was taken. The period of the plasmonic grating was chosen to be 580 nm, and the width of the air gap was 75 nm. The thickness of the metal was 80 nm, and the thickness of the ferrimagnetic film was larger, at 150 nm. The thicker layer of the magnetic layer is required for effective excitation of the waveguide modes [22].

The comparison of the transmission and the magneto-optical effect $\delta_{T}$ spectra in the nanostructures with uniform and modulated magnetization is given in Figure 6. The right column depicts the results for the magnetization modulation equal to P/2, i.e., to half of the period of the plasmonic grating.

Similarly, the SPP modes addressed the difference of the transmission spectra, which is negligible; however, the magneto-optical effect allows one to unambiguously detect the presence of excited spin waves. The typical nonzero magneto-optical effect accompanied by the similar sign of $\delta_{T}$ for opposite incidence angles of the input light is observed.

The spin wave detection in the spectral range of the waveguide modes acts in the same way as the SPP modes; however, it is preferable for the thick magnetic films where the SPP modes are distorted by the multiple waveguide modes. As soon as the same magnetoplasmonic nanostructure samples can support the excitation of both types of modes, the frequency range of the spin waves that can be detected by the proposed method is wide. Moreover, one sample could contain the set of plasmonic gratings with various parameters, namely, the period and air gap; therefore, scanning the magneto-optical effect from the different gratings one can detect the wide range of the spin waves.

## 5. Discussion and Conclusions

To sum up, the magnetization modulation created by the spin waves can be detected by means of the magnetoplasmonic nanostructures. We demonstrated that, in the presence of spin waves, the transverse MO intensity effect in transmission experiences a harmonic modulation and turns to zero with the periodicity equal to the period of the spin wave. This result can be used for the selective detection of the spin wave presence in the magnetic nanostructure. The proposed approach can also be extended to the near-IR spectral range as well as to other magneto-optical effects.

## Figures and Tables

**Figure 1 nanomaterials-12-00405-f001:**
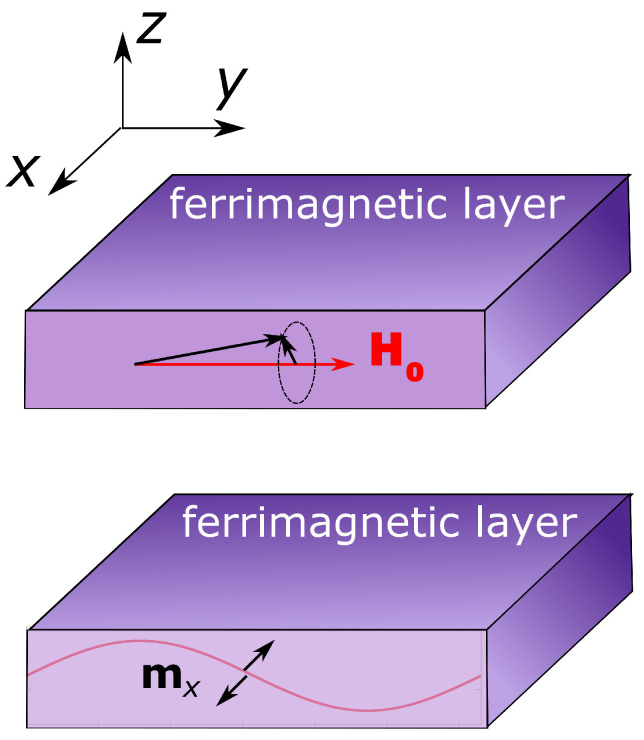
The scheme of the magnetization precession and the spatial dependence of the oscillating component of the magnetization $m_{x}$.

**Figure 2 nanomaterials-12-00405-f002:**
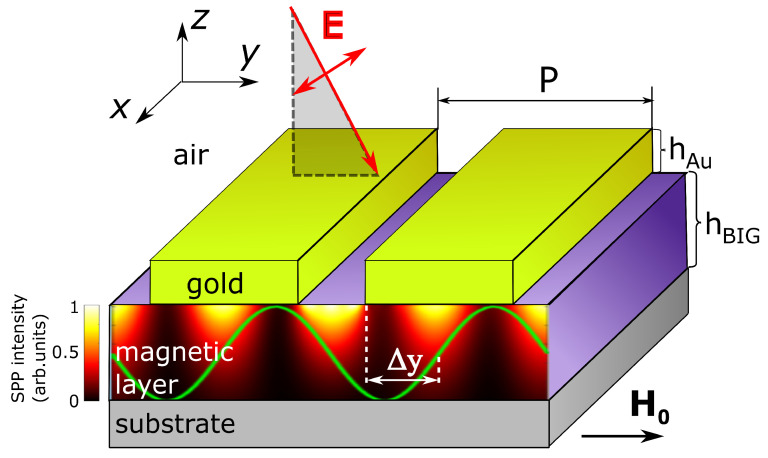
The scheme of the addressed magnetoplasmonic nanostructure. Gold grating is fabricated above the ferrimagnetic layer magnetized by the external magnetic field $H_{0}$ directed along the *y*-axis. The spin wave also propagating along the *y*-axis creates a modulation of the magnetization inside the magnetic layer shown by the green line. This leads to modulation of the gyration value $g_{x}$ along the *y*-axis (green curve). The nanostructure is illuminated by linearly *p*-polarized light (shown by the red oblique arrow) with the plane wavefront. The electric vector E of the input light lies in the plane zOy. The spatial distribution of the intensity of the optical field inside the ferrimagnetic layer related to the excitation of the SPP wave is shown by a yellow-red-black color plot. One can see that the period of the magnetization modulation is equal to double the period of the SPP wave. Parameter Δy is a spatial shift between the left edge of the nearest gold stripe (the start of the plasmonic grating period) and ‘zero’ of the gyration modulation. This parameter plays a role in the phase shift between the optical mode and the magnetization modulation. The parameters $h_{Au}$ and $h_{BIG}$ refer to the thickness of gold and ferrimagnetic layers, correspondingly.

**Figure 3 nanomaterials-12-00405-f003:**
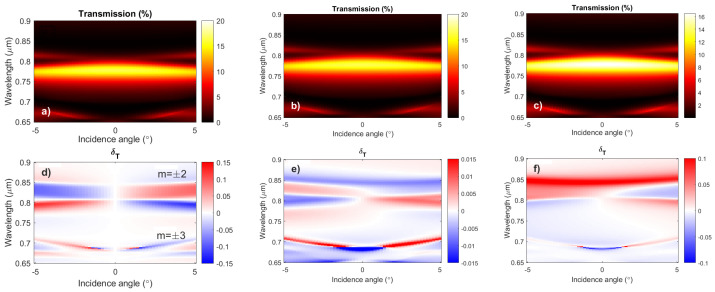
Angular and wavelength-resolved transmission (**a**–**c**) $\delta_{T}$ (**d**–**f**) spectra of the addressed nanostructure with the uniform spatial distribution of the magnetization in the ferrimagnetic layer (left column), magnetization modulation with $\lambda_{SW}=P$ (center column), and $2\lambda_{SW}=P$ (right column). The orders of the SPP modes in the left column are denoted by *m*. The phase shift between the plasmonic grating and the magnetization modulation is 0.

**Figure 4 nanomaterials-12-00405-f004:**
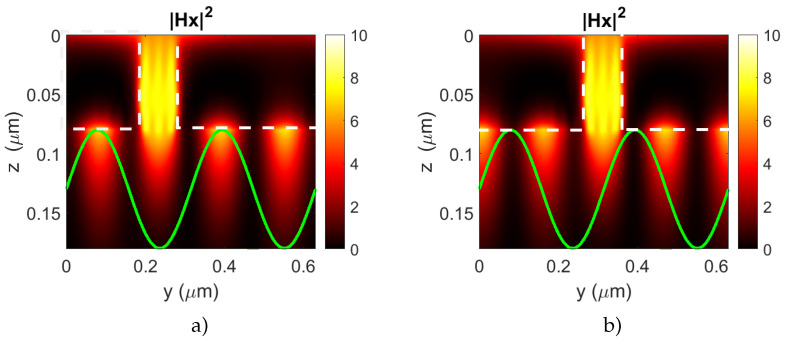
The spatial distributions of the $|H_{x}{|}^{2}$ of the SPP wave and the magnetization modulation (green line) for two different values of phase shift between the plasmonic grating and the magnetization modulation (**a**) ϕ=1.9 rad ($\delta_{T}=0$) and (**b**) ϕ=2.7 rad ($\delta_{T}=0.15$). The magnon wavelength is half of the plasmonic grating period. White dashed lines denote the cross sections of the gold stripes of the plasmonic grating.

**Figure 5 nanomaterials-12-00405-f005:**
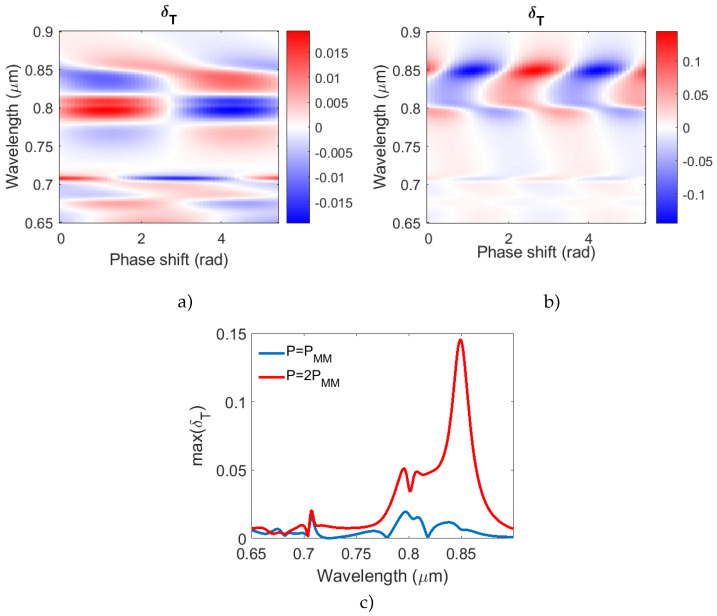
The magneto-optical effect $\delta_{T}$ versus the phase shift ϕ between the plasmonic grating and the spin wave for two periods of the magnetization modulation. (**a**) The magnetization modulation period is equal to the period of the plasmonic grating. (**b**) The magnetization modulation period is equal to the half of the period of the plasmonic grating. (**c**) The spectral dependence of the maximum value of $\delta_{T}$ for two different periods of magnetization modulation.

**Figure 6 nanomaterials-12-00405-f006:**
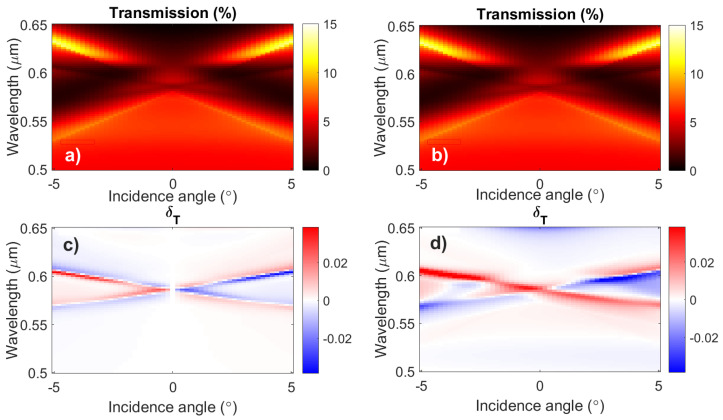
Angular and wavelength-resolved transmission (**a**,**b**) and $\delta_{T}$ (**c**,**d**) spectra of the nanostructure near the waveguide modes excitation area with (right column) and without (left column) the spatial modulation of the magnetization in the ferrimagnetic layer. The phase shift between the plasmonic grating and the magnetization modulation is 0.

## Data Availability

The data presented in this study are available within this article. Further inquiries could be directed to the authors.

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
