# Peer review of "Spectrally Selective Detection of Short Spin Waves in Magnetoplasmonic Nanostructures via the Magneto-Optical Intensity Effect"

_nanomaterials, 2022, doi:10.3390/nano12030405_

Round 1

Reviewer 1 Report

The manuscript presents an interesting approach for the observation of spjn waves in magneto-optically active materials based on the combination with magnetoplasmonic nanostructures. The authors should address some issues before the manuscript can be accepted for publication:

1. The wavelength (and frequency) of the spin waves should be constant for a given k vector. This implies that the relationship between the spin wave wavelength and the period of the plasmonic period should be determined by the nature of the material where the spin wave excitation takes place. The authors should show a plot of the dispersion of the spin wave wavelength as a function of photon wavelength in the investigated range for BIG and a transmission spectrum of BIG (for different incidence angles).

2. The authors should provide details on the software / code used for the simulations. Since not mentioned, I assume that all the results in the manuscript are results of theoretical simulations. Is that correct?

3. Figure 2: what represents the 2-D orange-red modulation? There is no relationship between this modulation and the period of the grating. Why? What does “input plane p-polarized light” mean? “incident linear p-polarized light”? Pleas explain all the quantities shown in the figure (delta y, H0, etc)

4. Line 132-133: the first part of the following sentence is incomplete and difficult to understand “So, 2n per period the phase shift is Ï• = 0 or Ï• = π” I can only guess that the authors refer to the case when kSWy = 0 or π. Anything else makes no sense. What is 2n per period?

5. Incidence angle is sometimes given in degrees, sometimes in rad. There should be a consistency.

Author Response

We thank the Referee for positive evaluation of our work. We have corrected the manuscript according to the comments given above.

  1. We thank the Referee for drawing our attention to this point that was unclear in the manuscript. We do not analyze the dispersion properties of the spin waves in the considered nanostructure because for the detection purposes we need to understand how spin waves change the magnetization distribution in the material and how the optical properties will change thereafter. But we include Refs. [30,31] where the dispersion properties of the corresponding materials are studied.

We provide the discussion of the magnetization and optical properties change due to the propagation of the spin waves in ferrimagnetic dielectric material to Section 2 as follows

‘Magnetization modulation in a ferrimagnetic dielectric material can be created in different ways. Among them there are the alternating magnetic domains [24,25], external mechanical stress or stretching [26,27], and excitation of spin waves [28,29]. In this paper we are interested in the latter option. When the spin waves propagate in magnetized ferrimagnetic dielectric medium, the magnetic dipole moments deviate from the direction given by the external magnetic field. As a result, the magnetization of the ferrimagnetic dielectric becomes modulated along certain direction. In this Section we consider how the oscillating components of the magnetization vector will be directed depending on the spatial coordinate. The further analysis will be based on the obtained spatial distribution of the magnetization vector.

<…>

The dispersion properties of the analyzed magnetic waves have been thoroughly studied in different magnetic materials, like yttrium iron garnet [30] or bismuth-substituted iron garnet [31]. So, it is necessary to understand how the propagating spin waves change the optical properties of the material via the magnetization variation.’

To stress the main idea of the manuscript we added to Section 3 the following discussion

‘If the wavelength of the spin wave is equal to integer number of the periods of the plasmonic grating, the MO effects observed in such nanostructure have some peculiarities that can be used for the detection of the magnons. We should emphasize that the described method won't detect all the spin waves propagating in the ferrimagnetic layer, but just the magnons of the certain wavelength. This causes the spectral selectivity of the proposed approach.’

  1. We are grateful to the Referee for this comment. We indicated the employed numerical method, rigorous coupled-wave analysis, and specified the idea of the method in the text in Section 4:

‘We have addressed the MO properties by means of the numerical simulation of the periodic magnetoplasmonic nanostructures by the rigorous coupled-wave analysis (RCWA) [39,40]. This method is an appropriate tool for simulation of optical and magneto-optical response of multilayered structures with lateral 1D or 2D periodicity. The first step of the method is the solution of the Maxwell’s equations in truncated Fourier space, so that Bloch waves are found as the solution of eigenwaves problem. Then at the second step the boundary conditions for interlayer interfaces are applied to obtain the algebraic set of equations for transmission and reflection coefficients. RCWA method provides calculation of both far-field characteristics (such as intensity and polarization of scattered waves for all diffraction orders) and near-field distribution of electromagnetic field components.’

  1. We added the description of the colour plot given in Fig. 2 and the explanation of the polarization state:

‘When the plasmonic nanostructure is illuminated by the plane wave of p-polarized light (the vector E lies in plane zOy as it is shown in Fig. 2), the SPP wave is excited [17]. Strictly speaking, there are excited two SPP modes propagating along y-axis forward and backward. The corresponding spatial intensity distribution of the SPP wave is shown in the ferrimagnetic layer by yellow-red-black color plot. It is given in the arbitrary units and is normalized to the peak value. Along with the optical light distribution the ferrimagnetic layer experiences the modulation of the dielectric properties due to the propagation of the spin wave and the related magnetization modulation.’

And in the capture of Fig. 2 we also provide the detailed description of the figure:

‘Gold grating is fabricated above the ferrimagnetic layer magnetized by the external magnetic field H0 directed along y-axis. The spin wave propagating also along y-axis creates a modulation of the magnetization inside the magnetic layer shown by the green line. This leads to modulation of the gyration value gx along the y-axis (green curve). The nanostructure is illuminated by linearly p-polarized light (shown by red oblique arrow) with plane wavefront. The electric vector E of input light lies in plane zOy. The spatial distribution of the intensity of optical field inside the ferrimagnetic layer related to the excitation of the SPP wave is shown by yellow-red-black color plot. One can see that the period of the magnetization modulation is equal to double period of the SPP wave. Parameter Δy is a spatial shift between the left edge of the nearest gold stripe (the start of the plasmonic grating period) and 'zero' of the gyration modulation. This parameter plays a role of the phase shift between the optical mode and the magnetization modulation. The parameters hAu and hBIG refer to the thickness of gold and ferrimagnetic layers, correspondingly.’

The period of the plasmonic grating and the wavelength of the SPP wave are connected by the condition of the equality of the wavevector components along y-axis. This condition is given for instance in refs. [17,36]. For the nanostructure addressed in the manuscript it gives the SPP wavelength of 313nm at the wavelength of 850nm at the normal incidence of light.

Fig. 2 was modified, the color bar was added.

  1. We agree with the Referee that the previous wording was confusing, so we clarified it as follows

‘From Eq. (3) one can see that MO effect depends on the phase shift φ, or, equally, on Δy. By varying the spatial shift Δy between plasmonic grating and magnetization modulation the MO effect can be controlled and even turn to zero, when kSWy+φ =0 or kSWy+φ =π. As it was mentioned above the plasmonic grating period should be equal to integer number of magnon wavelength, i.e. P=nλSW. It can be easily shown that the phase kSWy+φ takes the values 0 and π for 2n times in the interval [0,2πP/λSW). ’

And earlier on page 4:

‘Parameter Δy can take any values in the interval [0,P), where P is a period of the plasmonic grating. The right boundary of the interval is punctured, as it already belongs to the next period. Thus, the phase φ lays in the interval [0,2πP/λSW). Due to the periodicity of the magnetization in the ferrimagnetic layer, the total phase kSWy+φ changes inside the same interval [0,2πP/λSW).’

  1. We use the degrees for the angle of the input light, and we measure the phase shift φ in radians. We give some comments in the text of the manuscript to explain the choice.

On page 6:

‘There were calculated angular and wavelength-resolved transmission and MO effect spectra. Although the normal incidence of light is of the most interest in context of the asymmetric magnetoplasmonic nanostructures, the spectra are calculated also for oblique light as it is shown in Fig. 2 to facilitate the analysis and to make SPP resonances easily distinguishable. The input light angle is given in Fig. 3 in degrees and is measured from the normal to the sample plane.’

And further on page 8:

‘It is shown in Fig. 5, where the spectra of the MO effect versus the phase shift φ for two different wavelengths of the spin wave are given. The phase shift is given in radians to avoid confusing with input light angle measured in degrees earlier.’

Reviewer 2 Report

The work's topic is interesting, and the concept idea is good, but I suggest adding experimental results to support the numerical results. Only simulations are not sufficient for this kind of journal. I recommend another journal without both experimental and numerical results.

Author Response

We are grateful to the Referee for high estimation of the idea of our work.

When choosing a journal, we were guided by Aims & Scope of Nanomaterials (https://www.mdpi.com/journal/nanomaterials/about). In particular, it is stated there that ‘<…> theoretical and experimental articles will be accepted, along with articles that deal with the synthesis and use of nanomaterials. <…> Our aim is to encourage scientists to publish their experimental and theoretical research in as much detail as possible.’ So, we hope that due to the novelty of the proposed approach and its simplicity in comparison to the well-established the Brillouin light scattering (BLS) spectroscopy and μBLS technique, it will be interesting to the scientific community, the readers of Nanomaterials.

Reviewer 3 Report

The manuscript reports an effective method for the
spectroscopically selective detection of short spin waves via the
magneto-optical intensity effect in magnetoplasmonic
nanostructures with magnetization modulation. I have enjoyed
reading the manuscript and find the described method interesting
and surely worth of publication in MDPI Nanomaterials.

The paper is relatively well written, but I would suggest some
modifications and clarifications especially in the Abstract and
possibly in the Title. Specifically:

1. The Abstract starts with the phrase about "a novel method".

According to the policy of many journals, including Physical
Review series, Authors should not claim the novelty or priority of
their results, because it is quite difficult to quickly and
reliably verify the validity of such claims by referees and
editors. For example, "Physical Review adheres to the following
policy with respect to use of terms such as "new" or "novel:" All
material accepted for publication in the Physical Review is
expected to contain new results in physics. Phrases such as "new,"
"for the first time," etc., therefore should normally be
unnecessary; they are not in keeping with the journal's scientific
style. Furthermore, such phrases could be construed as claims of
priority, which the editors cannot assess and hence must rule out."
https://journals.aps.org/authors/new-novel-policy-physical-review

I am not sure whether the MDPI Nanomaterials Editors strictly
follow the same policy. If yes, I would suggest the Authors to
replace the above-mentioned phrase and similar statements in the
main article by a direct comparison of their results with some
known approximate solutions.

2. I think the main advantage of the described method compared to
the well-known ones (especially those based on measuring the
Faraday or Kerr rotation angles) is its spectral selectivity. This
advantage is explained in the section on "Discussion and
Conclusions", but, in my opinion, it is not adequately stressed in
the Abstract and it is not mentioned in the Title.

If the Authors agree with this opinion, I would suggest to modify
the article Title, e.g., as follows: "Selective detection of short
spin waves via the magneto-optical intensity effect".

3. I think for many readers, a more informative Title of the paper
could be, e.g.: "Selective detection of short spin waves in
magnetoplasmonic nanostructures with magnetization modulation."
But I do not insist on such a change of the Title.

Of course, even more informative title would be: "Selective
detection of short spin waves in magnetoplasmonic nanostructures
with magnetization modulation via the magneto-optical intensity
effect." But such a title might be considered too long.

* The Authors stress (including the Title and Abstract) that their
method is designed for detecting *short* spin waves. A reader
would expect a brief explanation in the Abstract why the method is
limited to detecting short waves. I assume that this range of
measured spins waves is related to a relatively long period of the
plasmonic grating. This could be explained in relation to the
Authors description in the main text that: "If the wavelength of
the spin wave corresponds to the period of the plasmonic grating
the MO effects observed in such nanostructure have some
peculiarities that can be used for the detection of the magnons."

Moreover, the wavelength range of such measured short spin waves
could be specified in the Abstract.

Author Response

We thank the Referee for his/her high evaluation of our work.

  1. We agree that the clarification ‘a novel method’ is redundant, the scientific data published by Nanomaterials should be novel and original due to the policy of the journal. Therefore, we removed ‘novel’ from the Abstract:

‘A method of spectrally selective detection of the short spin waves (or magnons) by means of the transverse magneto-optical (MO) intensity effect in transmission in the magnetoplasmonic nanostructure is proposed.’

  1. We thank the Referee for his/her proposals for the title of the manuscript. We have substituted the previous title by the following ‘Spectrally selective detection of short spin waves in magnetoplasmonic nanostructures via the magneto-optical intensity effect’ to stress the spectral selectivity of the proposed method that distinguishes the proposed method from existing ones.

  1. We changed the title of the manuscript to make it clearer for the readers.

Besides that, the discussion on the wavelength of the addressed spin waves have been added to the Abstract and Introduction.

We added the following clarification to the Abstract:

‘We considered the spin waves with the wavelength equal or less (by an integer number of times) of the period of the plasmonic structure, that is, of the order of hundreds of nanometers or 1-2μm.’

to the Introduction:

‘Thus, we are interested in the detection of the short spin waves, i.e. the spin waves with the wavelength from several hundreds of nanometers to 1-2μm.’

and to Section 3:

‘If the wavelength of the spin wave is equal to integer number of the period of the plasmonic grating, the MO effects observed in such nanostructure have some peculiarities that can be used for the detection of the magnons. We should emphasize that the described method won't detect all the spin waves propagating in the ferrimagnetic layer, but just the magnons of the certain wavelength. This causes the spectral selectivity of the proposed approach. Moreover, the requirement of the coincidence of the plasmonic grating and spin wave period means that the proposed method can be effectively employed for the detection of the short spin waves with the wavelength of less than 2μm. The greater periods of the plasmonic grating won't provide the effective excitation of the SPP waves and the detection scheme should be changed.’

Round 2

Reviewer 2 Report

The work can be accepted in its current form.